# Downregulation of Androgen Receptors upon Anabolic-Androgenic Steroids: A Cause or a Flawed Hypothesis of the Muscle-Building Plateau?

Heitor O. Santos [1],* and Carlos E. F. Haluch [2]

[1] School of Medicine, Federal University of Uberlândia (UFU), Uberlândia 2121, MG, Brazil
[2] Postgraduate Program, UNIGUAÇU, São Miguel do Iguaçu 84600-904, PR, Brazil; duduhaluch@yahoo.com.br
* Correspondence: heitoroliveirasantos@gmail.com

**Abstract:** Researchers and health practitioners seek to understand the upper limit of muscle hypertrophy under different conditions. Although there are models to estimate the muscle-building threshold in drug-free resistance training practitioners, little is known about the population using anabolic–androgenic steroids (AASs) in this regard. Because of a plateau effect of muscle hypertrophy upon AAS regimens, there is a hypothesis among clinicians and enthusiasts that AASs downregulate skeletal muscle androgen receptors (ARs). Conversely, in this narrative review, we show that seminal and recent evidence—primarily using testosterone and oxandrolone administration as human experimental models—support that AASs upregulate ARs, eliciting greater anabolic effects on skeletal muscle receptors through a dose-dependent relationship. Thus, to date, there is no scientific basis for claiming that myocyte AR downregulation is the cause of the AAS-induced plateau in muscle gains. This phenomenon is likely driven by the neutral nitrogen balance, but further research is imperative to clarify the intrinsic mechanisms related to this landscape.

**Keywords:** androgen receptor; anabolic–androgenic steroids; testosterone; skeletal muscle; muscle hypertrophy





## 1. Introduction

The use of anabolic–androgenic steroids (AASs) is targeted at clinical populations suffering from loss of strength and muscle mass [1–3]. Testosterone (T) administration is mainly used in the treatment of male hypogonadism as a means of improving a range of clinical symptoms and body composition [4–6].

T, as well as its synthetic derivatives (e.g., oxandrolone, stanozolol, oxymetholone, and nandrolone) can be used in muscle wasting disorders, such as burns, sarcopenia, osteoporosis, and cancer [7–12]. Notwithstanding the therapeutic proposals for AAS regimens, it is known that there is serious growth aimed at aesthetic improvement, within which muscle hypertrophy is one of the main goals [13]. Such a practice is not supported by the medical literature, with no consensus on dosing regimens and types of drugs, therefore resulting in an undecipherable field. However, due to the high prevalence of AAS use among non-athletes and athletes of different sports, as well as in resistance training practitioners [14,15], this scenario should not be neglected among health professionals who deal with these individuals.

Androgen receptors (ARs) are ligand-responsive transcription regulators regarded as a central tenet of muscle hypertrophy [16]. ARs are localized in the cytoplasm and translocate to the nucleus in the presence of AASs, thereby modulating AR-responsive genes [17]. There is a strong belief by health practitioners and enthusiasts that the constant AAS use at supraphysiological doses in bodybuilders and recreational AAS users can lead to a downregulation of ARs by virtue of putative saturation [18,19]. Such a hypothesis is

based on the concept that AAS-induced muscle gains reach a plateau after a certain period of AAS use, but this concept is still not well documented.

To the best of our knowledge, there is no narrative review to distinguish misunderstandings from pragmatic molecular and clinical insights in this regard; therefore, a critical appraisal of different areas of medical research is needed to draw better conclusions and future directions. That said, in this narrative review, we discuss the nuances of AR regulation over AAS use, focusing on lean body mass and related molecular parameters (satellite cells, myofibrillar protein synthesis, etc.) as well.

## 2. Downregulation of Hormone Receptors: Androgen Receptors vs. Other Receptors

Recognizably, many hormone receptors can be downregulated, such as adrenergic receptors (i.e., alpha 1-, alpha 2-, and beta-adrenergic receptors) and insulin receptors [20–24]. Chronic exposure to high levels of insulin and adiposity (i.e., obesity) leads to the downregulation of insulin receptors [25–27], while physical exercise and weight/fat loss not only upregulate these receptors and their substrates (insulin-receptor substrates 1 and 2) but also peroxisome proliferator-activated receptor-alpha (PPAR-$\alpha$) and -gamma (PPAR-$\gamma$), thus enhancing insulin sensibility and lipid metabolism [28–36].

Nevertheless, taking into account the relationship between ARs and skeletal muscle receptors, it is unsubstantiated to claim that there is downregulation of ARs in response to AAS administration. There is an undeniable lack of evidence to support AAS use downregulating ARs, even when considering animal studies. Amid the paucity of direct evidence, a single, older study using cells from the corpus cavernosum of rats reported that T led to downregulation of ARs [37]; however, instead of administrating T, finasteride was used as a means of increasing T concentration because of its actions in inhibiting 5-alpha reductase—then converting T to dihydrotestosterone [37]. In contrast to this study, another in vitro experiment showed that T administration upregulates AR in cultured the skeletal muscle satellite cells and myotubes of a porcine [38]. Thus, the latter study [38] portrays the biological nexus between AASs and AR regulation with more specificity in an attempt to understand the mechanisms inherent in the muscle-building plateau than the former one [37]. Furthermore, a couple of human studies show that T and oxandrolone upregulate myocyte AR [39,40], as discussed thoroughly in Section 4.

In addition to AASs, thermogenic drugs are widespread in bodybuilding and fitness [41,42]. Beta-2 adrenergic receptor agonists, such as clenbuterol and ephedrine, markedly promote the downregulation and desensitization of beta-2 receptors in the medium- and long-term, especially at high doses [43]. For this reason, these drugs lose effectiveness with chronic use this is quite different from long-term AAS use, which at least can maintain the muscle mass accretion induced by the initial and middle phases of AAS therapy [44,45]. As with AASs, it is crucial to emphasize that thermogenic drugs lack consensus as an approach to improving body composition due to adverse effects related to the cardiovascular system [46]—so much so, that they are not considered first-line therapy for obesity [47].

## 3. Crosstalk between Androgen Receptors and Satellite Cell-Mediated Hypertrophy

Muscle AR content has emerged as a determining factor for skeletal muscle hypertrophy, at least in men [48,49]. Albeit satellite cells are the predominant site of AR expression, ARs are expressed in various cell types in human skeletal muscle, such as fibroblasts, vascular endothelial, smooth muscle cells, mast cells, and CD34+ precursor cells [50]. Moreover, ARs are expressed in motor neurons, whose cells are located in the central nervous system and innervated to a target muscle, rendering a stimulus for myonuclear production due to the binding of AASs to ARs [51,52].

Satellite cells proliferate, differentiate, and fuse with each other, generating new myofibers with ensuing incorporation into an existing muscle fiber by donating their nucleus and yielding skeletal muscle hypertrophy [53]. The stimulus for the first step (i.e., proliferation) is triggered by resistance training or AAS use—or by their combination [54–57].

Many animal and human studies support that T administration leads to satellite cell proliferation and increases the number of myonuclei [58–60]. More importantly, supraphysiological doses of T increase the number of satellite cells and myonuclei in healthy men [60]. After 20 wk of 125, 300, or 600 mg weekly doses of T enanthate, Sinha-Hikim et al., detected significant increases in myonuclear number for those subjects who received 300 ($2.5 \pm 0.8$ to $5.0 \pm 0.8\%$, n = 8) and 600 mg ($2.5 \pm 0.5$ to $15.0 \pm 1.5\%$, n = 5) [60]. Equally important, the increase in satellite cell number correlated with changes in total (r = 0.548) and free T concentrations (r = 0.468) [60].

Given the well-established effects of AASs on orchestrating satellite cells, one can contemplate crosstalk between ARs and signaling pathways involved in activating satellite cells from quiescence to proliferation, as the binding of AASs to ARs is imperative to afford satellite cell-mediated hypertrophy [61].

Hence, based on the available literature, the hypothesis that the use of T and other AASs promote AR downregulation in skeletal muscle is an unwarranted assumption, as their effects on muscle mass accretion are evident, as discussed below.

## 4. Androgen Receptor Content, Accompanying Hypertrophy Mediators, and Lean Body Mass

Satellite cells and ARs regulate many genes in skeletal muscles. In a recent randomized clinical trial examining physically active men without obesity (n = 50) over 28 days of an exercise- and diet-induced 55% energy deficit, muscle AR protein and total RNA content were higher for those receiving 200 mg of T enanthate/wk than the placebo group while reducing fibroblast growth factor-inducible 14 and interleukin-6 receptor signaling [39]. In other words, T administration was able to attenuate proteolytic gene expression and enhance the translational capacity of myofibers, which collectively ensure greater myofibrillar protein synthesis and muscle hypertrophy. Not surprisingly, a lean body mass accretion of $3.8 \pm 1.2$ kg was found for the T group, with a reduction of $0.9 \pm 1.0$ kg for the placebo group ($p < 0.01$ between groups) [39].

The seminal study by Bhasin et al., tested triple the dose of the study above, observing that 600 mg/wk of T enanthate plus a resistance training program for 20 wk increased fat-free mass by ~6 kg ($65.3 \pm 1.8$ to $71.4 \pm 1.8$) in healthy men, whereas resistance training alone (placebo administration) showed an increase of 2 kg ($72.1 \pm 2.3$ to $74.1 \pm 2.2$) [62]. Additionally, in another study, Bhasin et al., demonstrated that T enanthate at 25, 50, 125, 300, or 600 mg for 20 wk (resulting in mean total T levels of 253, 306, 542, 1345, and 2370 ng/dL, respectively) increased fat-free mass in a dose-dependent manner when supraphysiological was administered (i.e., >100 mg/wk) in healthy young men (n = 61), with a fat-free mass accretion of 3.4, 5.2, and 7.9 kg for 125, 300, or 600 mg of T weekly, respectively [63].

Besides T administration, oxandrolone increases AR expression in skeletal muscle along with myofibrillar protein synthesis, as confirmed by a short-term study (6 healthy men) consisting of 15 mg/d oxandrolone for 5 days [40]. Such a daily dose is often given in a variety of clinical populations for which oxandrolone may be recommended [64–67]. Interestingly, Grunfeld et al., tested different doses of oxandrolone (20, 40, or 80 mg/d) for men with HIV-associated weight loss, and through a dose-dependent manner, found that only the 40 and 80 mg oxandrolone groups increased body weight and body cell mass compared to a placebo group over a 12 wk treatment period [68]. Body weight increased by $2.8 \pm 3.3$ and $2.3 \pm 2.9$ kg, and body cell mass increased by $1.5 \pm 2.5$ and $1.8 \pm 1.8$ kg, for 40 and 80 mg of oxandrolone, respectively, compared to their baselines.

These data are, therefore, a nail in the coffin for the unproven premise that myocyte AR are dose-dependently affected by AAS use. Indeed, insights into AAS dose-dependent muscle adaptations are proven by many studies [63,69–71], supporting a dose-dependent increase in skeletal muscle mass along with leg strength and power rather than AR downregulation and early muscle-building plateau in response to AASs. Not surprisingly, particular facets of muscle morphology are also dose-dependent, so athletes using a higher

AAS dosage (>2500 mg/wk) have larger muscle fiber areas than athletes using a "lower" AAS dosage (<500 mg/wk) and drug-free athletes [71]. Based on molecular results, greater muscle fiber nuclei and capillarization can be detected in doped athletes regardless of AAS dosage compared to drug-free athletes, hence favoring muscle hypertrophy and sports performance [71].

The sum of potential mechanisms for AR-mediated hypertrophy discussed in Sections 3 and 4 can be seen in Figure 1.

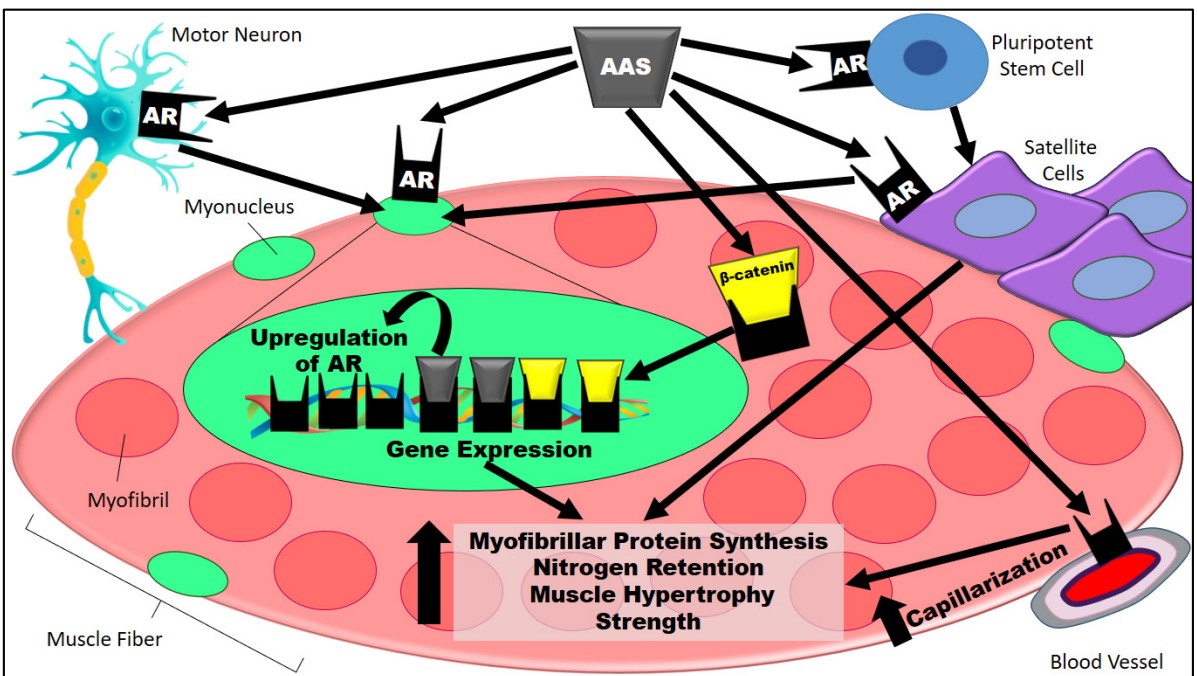

**Figure 1.** AAS-induced muscle hypertrophy is triggered by the binding of AASs to ARs in many muscle fiber-associated cells, such as motor neurons, pluripotent stem cells, satellite cells, and endothelial cells. AASs bind to ARs in pluripotent stem cells and, hence, originate in satellite cells [72]. In addition, AASs bind directly to satellite cells [73]. Satellite cells stimulate new myonuclear accretion [74]. Likewise, motor neurons stimulate myonuclei formation upon the binding of AASs to ARs [51,52]. In the myonucleus, ARs are upregulated by the presence of AASs [50]. ARs are capable of activating muscle hypertrophy-related gene expression without the nuclear presence of AASs, but the sarcoplasmic AAS-dependent stimulus of the AR/β-catenin complex is a fundamental process that affords the β-catenin translocation to the nucleus [75,76]. Additionally, the binding of AASs to ARs stimulates vascular cell proliferation via upregulation of vascular endothelial growth factor A and cyclins (e.g., cyclin A and D1), thus entailing capillarization and consequently improving the influx of nutrients as well as the supply of oxygen and growth factors to skeletal muscle [50,77,78]. Taken together, the aforementioned pathways elicit greater myofibrillar protein synthesis and nitrogen retention, thereby translating into muscle hypertrophy and strength [79]. AAS, anabolic–androgenic steroid; AASs, anabolic–androgenic steroids; AR, androgen receptors.

## 5. Why Can Downregulation of Myocyte Androgen Receptors in Response to Anabolic–Androgenic Steroids Be Considered a Flawed Hypothesis?

Taking into consideration the real bodybuilding scenario, i.e., competition without doping tests, the loss of muscle mass under high doses of T and associated AASs with high anabolic potential (e.g., nandrolone and oxymetholone) is almost illogical despite the assumed threshold for skeletal muscle gains. The same is true for clinical populations (female-to-male transgender persons, patients with HIV/AIDS, sarcopenia, etc.), whose muscle anabolism is the expected effect induced by AAS regimens, seemingly reaching a plateau with the maintenance of a therapeutic dosage [80–84]. In these cases, the muscle-building plateau must not be considered an AR-disrupting factor, as if the chronic use of T

and general AASs promoted the downregulation of ARs, muscle mass gains would not be sustainable irrespective of the population.

If there was downregulation of ARs upon AAS administration, T and its derivatives should cease to be effective over time in order to avoid a putative reduction in myofibrillar protein synthesis and muscle mass. On the other hand, there is considerable muscle mass loss after ceasing AAS use—particularly for those agents with greater anabolic potential [85]—due to a negative nitrogen balance (i.e., less muscle protein synthesis and more protein breakdown) [86–88].

## 6. AAS Misuse vs. Therapeutic Use

Although AAS users (i.e., recreational exercise practitioners) are prone to practice polypharmacy, such as combining AASs with diuretics, stimulant thermogenic substances (ephedrine, clenbuterol, caffeine, etc.), and illegal psychotropic substances (e.g., cocaine) [42,89,90], there is an alarming amount of caution geared toward AAS abuse. In light of this, the dose-dependent effect of AASs on pathophysiological responses must be considered. In a survey consisting of 500 AAS users (78% noncompetitive bodybuilders and non-athletes), ~60% reported using at least 1000 mg of T or its derivatives per week [91]. AAS misuse of this magnitude ought to be considered detrimental to health and therefore prohibited, as the traditional dosing regimen of TRT tends to be ~100 mg weekly or ~200 mg every 2 weeks (at least intramuscularly) [92].

Recreational AAS use is associated with many side effects, such as low gonadotropin and T levels, infertility, irritability, acne, and unfavorable liver and cardiovascular profiles, among others [19,93,94]. Viewed collectively, AAS abuse is associated with a harmful cardiometabolic profile due to increasing blood pressure as well as circulating levels of low-density lipoprotein cholesterol and total cholesterol while decreasing high-density lipoprotein cholesterol [95]. Correspondingly, sudden cardiac death in AAS users is associated with cardiomegaly and left ventricular hypertrophy, accompanied by fibrosis and necrosis of myocardial tissue, as well as atherosclerosis, inflammatory infiltrate, and coronary stenosis [96]. These macroscopic and histological alterations contribute to the intertwined link between pathophysiological cardiac remodeling and life-threatening arrhythmias [96].

On the other hand, recent evidence supports TRT as a safe and effective strategy for male hypogonadism [97–99]. Moreover, proper dosing regimens of certain AASs (e.g., oxandrolone, nandrolone, and stanozolol) may aid in mitigating muscle wasting disorders [7–12].

## 7. Take Home-Messages and Perspectives

Recent and seminal evidence supports that AASs—at least using T and oxandrolone administration in human experimental models—upregulate myocyte ARs, conferring greater anabolic effects on skeletal muscles [38–40]. Although there is a lack of compelling research, the data are in line with well-controlled, randomized clinical trials that employed different AAS dosages, since AAS-induced muscle gains are indisputable [62,63,68].

To date, the supposed hypothesis of AR downregulation caused by AAS use is a mere anecdote that cannot be translated into the real-world scenario of muscle hypertrophy because, to the best of our knowledge, only a former study published nearly three decades ago suggests T-induced downregulation of ARs using rat cavernosum smooth muscle, in which finasteride was used to increase T levels instead of T administration [37].

Since AASs upregulate ARs, further studies are warranted to portray the underlying causes of the plateau effect on muscle hypertrophy among the population for which AAS use is prevalent, i.e., bodybuilders, weightlifters, and individuals suffering from muscle wasting disorders. Conversely, it is worth mentioning that AASs are prescription-only medicines that are often used without medical advice to augment muscle mass and enhance sports performance; therefore, health practitioners should counsel patients about the detrimental effects of AAS abuse.

With this caveat in mind, the illegal practice of AAS use should not be endorsed or replicated (i.e., tested) in the spheres of human science; however, further animal research focusing on the effects of different AAS dosing regimens is essential to elucidate AR modulation and related mechanistic aspects of the muscle-building plateau, followed by well-controlled, randomized clinical trials addressing common AAS dosages on tight safety control. Regarding the latter, such a landscape could be explored in clinical populations using AAS dosing regimens approved for muscle wasting disorders, as scientists are endeavoring to understand the upper limit of muscle hypertrophy under different conditions.

**Author Contributions:** H.O.S.: conceptualization, investigation, supervision, writing—original draft, review, and editing. C.E.F.H.: investigation and writing—original draft. All authors have read and agreed to the published version of the manuscript.

**Funding:** H.O.S. has been supported by the Coordenação de Aperfeiçoamento de Pessoal de Nível Superior—Brazil (CAPES).

**Conflicts of Interest:** The authors declare no conflict of interest.

## Abbreviations

AASs, anabolic–androgenic steroids; AR, androgen receptors (AR); PPAR-$\alpha$, peroxisome proliferator-activated receptor-alpha; PPAR-$\gamma$, peroxisome proliferator-activated receptor-gamma; T, Testosterone.

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
