# Peer review of "Downregulation of Androgen Receptors upon Anabolic-Androgenic Steroids: A Cause or a Flawed Hypothesis of the Muscle-Building Plateau?"

_muscles, doi:10.3390/muscles1020010_

Round 1

Reviewer 1 Report

Manuscript “Downregulation of androgen receptors upon anabolic-androgenic steroids: A cause or a 2 flawed hypothesis of the muscle-building plateau?” is a review article devoted to the anabolic-androgenic steroids and mechanisms for the androgen receptors regulation.

Main remarks:

1.    What the main difference between this review and others devoted to the androgen receptors and anabolic-androgenic steroids? This point should be clearly mentioned.

2.    I see motoneurons at Figure 1 scheme. But no information exists in the main text. This information should be included. Unfortunately, no information about the neuronal regulation exists.

Author Response

Reviewer 1

Comments and Suggestions for Authors

Manuscript “Downregulation of androgen receptors upon anabolic-androgenic steroids: A cause or a 2 flawed hypothesis of the muscle-building plateau?” is a review article devoted to the anabolic-androgenic steroids and mechanisms for the androgen receptors regulation.

Main remarks:

  1. What the main difference between this review and others devoted to the androgen receptors and anabolic-androgenic steroids? This point should be clearly mentioned.

Response: Dear reviewer, thank you for suggestions. We have rewritten the last paragraph of the introduction to address the novelty of this paper. To the best of our knowledge, there is no critical review of the literature to distinguish misunderstandings from pragmatic molecular and clinical insights in this context, as a unifying link between these facets is needed to draw better conclusions and future directions.

  1. I see motoneurons at Figure 1 scheme. But no information exists in the main text. This information should be included. Unfortunately, no information about the neuronal regulation exists.

Response: We have cited information about motor neurons in the main text. Please see section 3 “…Moreover, AR are expressed in motor neurons, whose cells are located in the central nervous system and target muscle, rendering a stimulus for myonuclear production due to the binding of AAS to AR”. Thank you very much for showing us this part.

Reviewer 2 Report

Although the general topic is interesting, there are some concerns that should be highlighted:

(1) The study justification does not seem plausible, since the authors are justifying the importance of the current study based on the irresponsibility of athletes and users without medical prescription. In view of this, this study may contribute to the irresponsible use of anabolic androgenic steroids.

(2) Another point was that the authors pointed out only positive points, ignoring the side effects that are already well elucidated in the literature.

(3) The language must be improved; 

(4) There is inconsistences of described topic and its respective cited references: e.g., REF 69

(5) Most of cited studies are published long time ago; in this hand, this point is awkward, since nowadays athletes and users currently use dosages much higher than those of the cited studies.

Author Response

Reviewer 2

Comments and Suggestions for Authors

Although the general topic is interesting, there are some concerns that should be highlighted:

(1) The study justification does not seem plausible, since the authors are justifying the importance of the current study based on the irresponsibility of athletes and users without medical prescription. In view of this, this study may contribute to the irresponsible use of anabolic androgenic steroids.

  1. Response: Dear reviewer, thank you for this observation. This study does not provide any recommendation for anabolic-androgenic steroids without a medical prescription, mainly for aesthetic proposals. Instead, we have reinforced the side-effects by creating a new topic, please, check topic 6. “AAS misuse vs. therapeutic use”. Prior to this topic, we already discussed cautions against illegal use of AAS both in the introduction and in the last topic.

The focus of this research is to discuss the mechanistic link between the modulation of androgen receptors upon anabolic-androgenic steroids without any stimulus for improper use. We believe that the subject matter of this review is of great importance to the literature in order to understand the muscle-building plateau and the role of androgen receptors, which is currently a therapeutic target for many ailments.

(2) Another point was that the authors pointed out only positive points, ignoring the side effects that are already well elucidated in the literature.

Response: We have mentioned side effects in this current version with more emphasis. Undeniably, our scientific writing does not support the illegal use of steroids and we endorse caution as to the use. We only avoided discussing the side effects thoroughly in the previous version so as not to focus on other areas. However, we agree with you and believe it is crucial to highlight this context in detail.  Thus, as above-mentioned, we have created Topic 6. AAS misuse vs. therapeutic use”.

(3) The language must be improved; 

Response: We have checked the language throughout the paper. Even so, if you notice some syntax and specific grammar problems, please let us know. Fortunately, the MDPI system has an efficient team for final standardization, but we always try our best and are available for suggestions.

(4) There is inconsistences of described topic and its respective cited references: e.g., REF 69

Response: We have updated the reference list. Previous reference 69 is current reference 71. This is a review consisting of a robust background of mechanisms and clinical cases. Perhaps this was the main reference that furnished evidence for the creation of our figure. We are available for further reference updates if you deem it necessary.

(5) Most of cited studies are published long time ago; in this hand, this point is awkward, since nowadays athletes and users currently use dosages much higher than those of the cited studies.

Response: Indeed, most of the cited studies were published a long time ago. This occurred because interventions with high doses of steroids were carried out some decades ago, so the number of publications (RCTs) has decreased to avoid unnecessary exposure of patients to side effects. However, we believe it is crucial for our review in terms of unifying old studies with the more recent ones, including a mix of animal models and molecular research.

We agree that athletes and users currently use dosages much higher than those in the cited studies. However, it is not possible to associate this population with the mechanistic links discussed. Generally, this population is studied in an observational fashion focusing on side effects. At best, we have cited some cases of heavy use of steroids to show that it is impossible to establish a muscle-building plateau in virtue of the lack of control and unfeasible exposure to severe side effects. 

Reviewer 3 Report

This article wrote nicely about the use of AASs for people who lost muscle mass due to several diseases. as described in last paragraph, animal experiments should be made to get scientific knowledge.

Author Response

Reviewer 3

Comments and Suggestions for Authors

This article wrote nicely about the use of AASs for people who lost muscle mass due to several diseases. as described in last paragraph, animal experiments should be made to get scientific knowledge.

Response: Thank you very much for your kind comments. We agree with you that animal experiments should be done to gain scientific knowledge. In fact, it is unethical to expose humans to steroid use without a specific disease. We maintained this message in the last topic.

Round 2

Reviewer 1 Report

Manuscript can be accepted without answer to reviewer.

Author Response

Reviewer 1

Comments and Suggestions for Authors

Manuscript can be accepted without answer to reviewer.

Response: Dear reviewer, thank you so much for accepting our paper.

Reviewer 2 Report

I keep my first decision about rejection. The topic is interesting but the manuscript highlight a wrong way about anabolic steroids.

It is sad to know that the journal support this idea.

Author Response

Reviewer 2

Comments and Suggestions for Authors

I keep my first decision about rejection. The topic is interesting but the manuscript highlight a wrong way about anabolic steroids.

It is sad to know that the journal support this idea.

Response:

Dear editor, this reviewer was concerned about anabolic steroid misuse. As we already responded to him/her in the first round, we not only have cited the concern about steroid misuse since the first version (i.e., previous to the round 1), but also we have reinforced this by creating a specific topic in addition beyond to mention in the introduction. Unfortunately, this practice occurs in the real-world scenario and is a concern worldwide. No wonder we have mentioned the side effects and guidelines position.

Please, check this response in the first round (as well as the others): “Dear reviewer, thank you for this observation. This study does not provide any recommendation for anabolic-androgenic steroids without a medical prescription, mainly for aesthetic proposals. Instead, we have reinforced the side-effects by creating a new topic, please, check topic 6. “AAS misuse vs. therapeutic use”. Prior to this topic, we already discussed cautions against illegal use of AAS both in the introduction and in the last topic.”

We appreciate it when the reviewer says “The topic is interesting”. However, when the reviewer affirms “but the manuscript highlight a wrong way about anabolic steroids. It is sad to know that the journal support this idea.”, we think that he/her lacks specificity about the area.

As with any other drug or unhealthy agents, such as cannabis, alcohol, cigarettes, stimulants, etc., anabolic steroid misuse must be addressed in the literature in different ways. Side effects should be addressed such as problems related to the cardiovascular system, liver and central nervous system, tendon and muscle injuries, etc. In addition, the molecular aspects related to muscle must also be discussed, including those pathways considered favorable to hypertrophy. That is why we aimed at Muscles (MDPI journal).

We do not agree when the reviewer says that “It is sad to know that the journal support this idea”. In our view, Muscles is a vast journal and can address pharmacological agents, nutrition and sports strategies in modulating the muscular system.

Sports (MDPI journal) is an example of a similar journal that has gained a huge audience. There are many publications focusing on bodybuilders in the Current Issue of this journal, which were carried by recognized authors that would appreciate our paper, for example, the American Professors Grant Tinsley, Brad Schoenfeld, Guillermo Escalante, etc.

In addition, we communicate that our team recently published a paper about steroids (testosterone) in an MDPI journal (URO) which has been appreciated by medical researchers. Please, see:

Exploring the Role of Testosterone Replacement Therapy in Benign Prostatic Hyperplasia and Prostate Cancer: A Review of Safety https://www.mdpi.com/2673-4397/2/1/5

Also, the corresponding author of this paper would like to communicate that he has published many recent studies on steroids and non-steroid hormones in different areas of medical research and has a full conscience of his scientific reputation. See some examples below:

PMID: 32446600 PMID: 32304719 PMID: 34116112 PMID: 32745490 PMID: 32675010 PMID: 30183993 PMID: 34462124 PMID: 33080320 PMID: 31276773 PMID: 30767598 PMID: 30790614

There is no evidence of encouraging drug misuse in these articles. Therefore, we reinforce that the authors are skeptical and do not encourage the misuse of drugs; on the contrary, their research focuses on the security profile.

Notwithstanding the prejudgment by this reviewer, we are clean and transparent authors and we are available for further clarifications if necessary.